# Impacts of Land Cover Change on the Spatial Distribution of Nonpoint Source Pollution Based on SWAT Model

Zeshu Zhang [1,*], Hubert Montas [1], Adel Shirmohammadi [2,*], Paul T. Leisnham [2] and Masoud Negahban-Azar [2]

1   Fischell Department of Bioengineering, University of Maryland, College Park, MD 20742, USA
2   Department of Environmental Science and Technology, University of Maryland, College Park, MD 20742, USA
*   Correspondence: zhzeshu@umd.edu (Z.Z.); ashirmo@umd.edu (A.S.)

**Abstract:** Nonpoint source (NPS) pollution is a pressing issue worldwide, especially in the Chesapeake Bay, where sediment, nitrogen (N), and phosphorus (P) are the most critical water quality concerns. Despite significant efforts by federal, state, and local governments, the improvement in water quality has been limited. Investigating the spatial distribution of NPS hotspots can help understand NPS pollutant output and guide control measures. We hypothesize that as land cover changes from natural (e.g., forestland) and agricultural to suburban and ultra-urban, the distribution of NPS pollution source areas becomes increasingly spatially uniform. To test this hypothesis, we analyzed three real watersheds with varying land cover (Greensboro watershed for agriculture, Watts Branch watershed for suburban, and Watershed 263 for ultra-urban) and three synthetic watersheds developed based on the Watts Branch watershed, which ranged from forested and agricultural to ultra-urban but had the same soil, slope, and weather conditions. The Soil and Water Assessment Tool (SWAT) was selected as a phenomenological model for the analysis, and SWAT-CUP was used for model calibration and validation. The hydrologic responses of the three real and synthetic watersheds were simulated over ten years (1993–2002 or 2002–2011), and calibration and validation results indicated that SWAT could properly predict the export of runoff and three target NPS pollution constituents (sediment, total nitrogen, and total phosphorus). The results showed that the distribution of NPS pollutant outputs becomes increasingly uniform as land cover changes from agriculture to ultra-urban across watersheds. This research suggests that the spatial distribution of NPS pollution source areas is a function of the major land cover category of study watersheds, and control strategies should be adapted accordingly. If NPS pollution is distributed unevenly across a watershed, hotspot areas output a disproportionate amount of pollution and require more targeted and intensive control measures. Conversely, if the distribution of NPS pollution is more uniform across a watershed, the control strategies need to be more widespread and encompass a larger area.

**Keywords:** Nonpoint Source Pollution (NPS); SWAT; spatial distribution

## 1. Introduction

Nonpoint source (NPS) pollutants such as sediment and nutrients (nitrogen and phosphorus) are the most significant threat to water quality in the U.S. For example, the Environmental Protection Agency (EPA) estimated that urban runoff and storm sewer outflow were the fourth leading source of impairment in rivers, third in lakes, and second in estuaries [1]. The Chesapeake Bay, the largest estuary in the U.S., has suffered from excessive NPS pollutants in recent decades [2]. Anthropogenic activities by over 17 million people in this watershed have brought excess amounts of sediment and nutrients, which has made the bay highly eutrophic, reduced habitat for marine species, and led to economic losses to the seafood industry, boat charters, and tourism [3–7]. Therefore, research about controlling and managing NPS pollutants is crucially significant for water resource protection [8].

Humans and nature play a significant role in NPS pollutants' output, and managing NPS pollutants is a complex problem [5,9,10]. In general, it is more cost-effective to control pollution output in highly polluted areas, commonly called "hotspots" or "critical source areas (CSAs)" [11–13], where pollutants' output density is higher than in other areas. Best management practices (BMPs) are suggested to be implemented in pollution hotspots. The effectiveness of this method depends on the assumption that pollutants are not evenly distributed, and some hotspots will account for a higher percentage of pollutants' output than the other areas. It has been found that pollutants' output is relatively localized in natural (e.g., forestland) and agricultural areas [14,15]. Still, the correctness of this assumption in urban areas, especially highly urbanized areas, is unknown. Therefore, to guide BMP selection and implementation strategies in urbanized areas, finding the spatial distributions of NPS pollutants and identifying highly (i.e., critical) polluted areas is of considerable basic science and applied importance.

Hydrologic models have been developed to simulate the watershed response to various climatic and storm properties, analyze the pollutants' loadings, and guide control measures [16,17]. The structure of hydrologic models varies from simple lumped models that consider a watershed as a single land cover and management unit such as CREAMS (Chemical, Runoff, and Erosion from Agricultural Management Systems) [18] and N-SPECT (Nonpoint Source Pollution and Erosion Comparison Tool) [19], medium complex model LSPC (Loading Simulation Program in C++) [20], to complex spatially distributed models such as SWAT (Soil and Water Assessment Tool) [21–24], SWMM (EPA Storm Wate Management Model) [25]. Statistical and machine learning methods also have been used to model hydrologic processes. For example, the rainfall-runoff model based on the sequence-to-sequence LSTM model structure has shown its accuracy in hourly runoff prediction [26].

Model selection is a complex task, and many factors need to be considered, such as data availability, watershed properties, project costs, knowledge and skills, model complexity, processes needed to be simulated, etc. [27]. As distributed hydrologic models consider the spatial variability of hydrologic processes, it is generally more accurate to analyze the spatial characteristics of watersheds and guide BMPs as appropriate. Many research studies have demonstrated that SWAT can simulate hydrologic processes adequately in natural and agricultural watersheds, but the application of SWAT in urban areas is much less common [11,14,21,27–32]. Applying SWAT in both natural and urbanized areas potentially would provide vital comparisons of the spatial distribution patterns of NPS pollutants across diverse land covers and inform land cover management decisions [33].

This research aims to test the application of SWAT in urban areas, analyze the spatial distribution pattern of NPS pollutants, and identify highly polluted areas (i.e., hotspots/CSAs) among different land cover types. The research objectives were divided into the following steps: (1) develop SWAT model's input parameters reflective of three watersheds in the Chesapeake Bay drainage basin that varied with land cover (agricultural, suburban, ultra-urban); (2) identify the spatial distribution pattern of NPS pollutants among these three watersheds; and (3) confirm the spatial distribution patterns by SWAT model in three synthetic watersheds that varied with land cover, from forested and agricultural to ultra-urban, but were controlled for soil, slope, and weather, as synthetic watersheds allow for the evaluation of different land covers in a well-defined environment.

## 2. Materials and Methods

### 2.1. Watershed Description, Software, and Data Availability

This research concentrated on three watersheds within the Chesapeake Bay basin. They are the agricultural Greensboro watershed (Figure 1a), suburban Watts Branch watershed (Figure 1b), and highly urban Watershed 263 (Figure 1c). The location, land cover, soil, and slope of these three watersheds are shown in Figures 1 and 2.

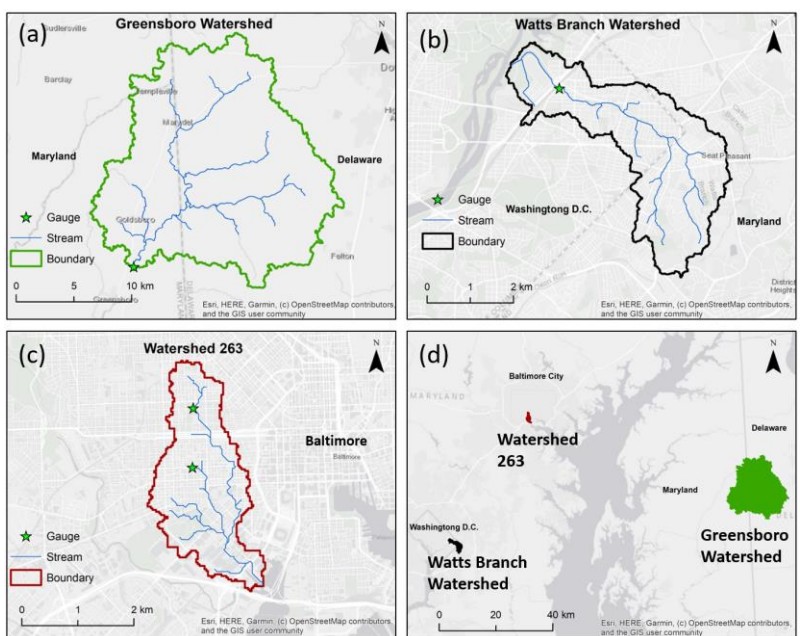

**Figure 1.** Location of the three study watersheds: (**a**) agricultural Greensboro watershed between Maryland and Delaware; (**b**) suburban Watts Branch watershed between Washington, D.C., and Maryland; (**c**) ultra-urban Watershed 263 in Baltimore city; and (**d**) relative position of three watersheds.

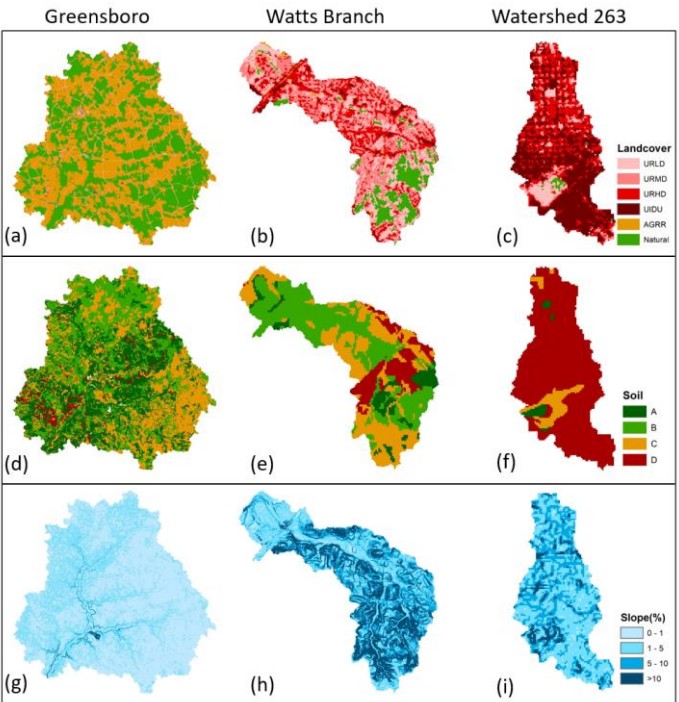

**Figure 2.** (**a–c**) Land cover, (**d–f**) soil, and (**g–i**) slopes of the study watersheds (Greensboro, Watts Branch, and Watershed 263).

The agricultural Greensboro watershed (Figure 1a) is a 298.27 km² subwatershed of the Choptank River watershed (USGS HUC 02060005) on the eastern shore of Maryland. The land use of the watershed includes cropland (48.9%), forested wetlands (32%), deciduous forests (7.6%), and urban land (5.7%). It has a flat topography with a mean slope below 1%. The major hydrologic soil groups are soil groups A (14%), B (38%), and C (46%). The suburban watershed considered in our research is the Watts Branch watershed (Figure 1b), which stretches between Maryland and Washington, D.C. The watershed occupies 10.4 km²,

a suburban area, with 22.1% low-density residential area, 37.2% medium-density residential area, 24.2% high-density residential area, and 12.9 % forest. The average slope is 7.8%, which is much higher than the Greensboro watershed. The third watershed considered in this research is Watershed 263 (Figure 1c), which is entirely urbanized and located in Baltimore, Maryland. The watershed covers 4.63 km$^2$, with 51.2% industrial urban area and 33% high-density residential area. The watershed is flatter than the Watts Branch but steeper than the Greensboro watershed, with an average slope of 5.3%. Soil group D accounts for more than 80% of the watershed.

The software, spatial, and time-series data used in the study are summarized in Table 1.

**Table 1.** Software and data used in the study.

| | Greensboro | Watts Branch | Watershed 263 |
|---|---|---|---|
| Software | | | |
| Model Development | ArcSWAT | ArcSWAT | ArcSWAT |
| Spatial Analysis | ArcGIS | ArcGIS | ArcGIS |
| Model Calibration | SWAT-CUP | SWAT-CUP | SWAT-CUP |
| Spatial Data | | | |
| Topography | USGS NED 2006 30 m DEM | USGS NED 2006 30 m DEM | USGS NED 2006 30 m DEM |
| Land Use/Land Cover | NLCD 2006 30 m shapefile | NLCD 2006 30 m Shapefile | NLCD 2006 30 m Shapefile |
| Soils | SSRUGO 2012 1:24,000 shapefile | SSURGO 2012 1:24,000 Shapefile | SSURGO 2012 1:24,000 Shapefile |
| Time-series Data | | | |
| Discharge (SurfQ) | USGS NO. 1,491,000, Points: 3651 | USGS N0. 1,651,800, Points: 3651 | NA |
| Total Sediment (Sed) | USGS NO. 1,491,000, Points: 197 | Points: 8 daily grab samples | NA |
| Total Nitrogen (Tot N) | USGS NO. 1,491,000, Points: 224 | Points: 8 daily grab samples | Points: 120 daily grab samples |
| Total Phosphorus (Tot P) | USGS NO. 1,491,000, Points: 217 | Points: 8 daily grab samples | Points: 120 daily grab samples |
| Weather Data | TAMU NOAA NCDC Stations: Precipitation, temperature, relative humidity, solar radiation, wind speed | Washington National Airport, VA, USW00013743, precipitation, temperature | Maryland Science Center, MD, USW00093784, precipitation, temperature |

ArcSWAT, an ArcGIS-ArcView extension, is used for the model's input parameter development. SWAT is a semi-distributed physical-based hydrologic model developed by the U.S. Department of Agriculture and Agricultural Research Service (USDA-ARS) [24]. SWAT operates in continuous time steps and has been widely used to simulate water quantity and quality in gauged and ungauged watersheds [22,23]. In the SWAT model, a watershed is delineated into several sub-watersheds. Sub-watersheds are further divided into hydrologic response units (HRUs) with the same land use, soil type, slope, and management practice. The hydrologic simulation output is based on the HRUs response. Other software includes ArcGIS for data preparation and spatial analysis and SWAT-CUP for model calibration and validation.

Spatial data about topography, land use, and soils were obtained from the United States Geological Survey (USGS), the United States Department of Agriculture-Natural Resources Conservation Service (USDA-NRCS), and the National Land Cover Database (NLCD). These data were directly downloaded from public websites. A coarse resolution of data could make the model faster to run, but it can also result in a loss of detail and precision. Thus, the selection of data should balance the computational efficiency and the model accuracy. The topography and land use data resolution is 30 m × 30 m, compromising model accuracy and computation speed. SSURGO soil data instead of STASGO were chosen for model input parameter development as SSURGO provides more detailed soil data in the watersheds.

Time-series data used for model development include stream discharge (SurfQ), total sediment (Sed), total nitrogen (Tot N), total phosphorus (Tot P), and weather data. SurfQ, Sed, Tot N, and Tot P were obtained from the USGS gauging stations from 2002 to 2011 in the Watts Branch watershed (USGS No. 1,651,800) and from 1993 to 2002 in the Greensboro watershed (NSGS No. 1,491,000). For Watershed 263, since there is no gauging station in the watershed, we obtained the nitrogen and phosphorus data from the Cary Institute of

Ecosystem Studies between 2004 and 2010. The SWAT model was calibrated and validated under the available data.

The input weather data of SWAT include precipitation, maximum and minimum temperature, solar radiation, relative humidity, and wind speed. These can be simulated or observed from the weather stations. For Watershed 263 and Watts Branch watershed research, the weather data were obtained from NOAA National Climate Data Center (USW00093784 and USW00013743). The dataset contains daily temperature and precipitation. The weather generator program (CLIGEN) embedded in the SWAT database generated the other three unavailable data [34]. For the Greensboro watershed, weather data were obtained from the NOAA National Weather Service (NWS) National Center for Environmental Prediction (NCEP) Climate Forecast System Reanalysis (CFSR) database.

### 2.2. SWAT Model Parameter Development in Three Watersheds

First, ArcGIS preprocessed the spatial data (topography, land use, soils), such as clipping the grid data to the appropriate extent, merging the clipped grids, and projecting the grid in a suitable UTM zone. Then ArcSWAT was used to combine them with time-series data to build the SWAT running files. This process identified the watershed and subwatershed boundaries depending on the topographic maps. Land use, soil, and slope threshold values were set to zero to build the smallest HRUs, consisting of homogeneous land use, soil, and slope characteristics [35]. It is the smallest land research unit for hydro process calculation. SWAT model options were run based on these input files.

SWAT is a widely used watershed-scale model that simulates the impacts of land management practices and climate change on water, sediment, and nutrient fluxes in a river basin. The model output is divided into two groups of variables: in-stream variables and on-land variables. On-land variables include surface runoff, sediment (t/ha), total nitrogen (kg/ha), and total phosphorus (kg/ha). These variables represent the spatial distribution of nonpoint source (NPS) pollutants on land. They are important for identifying areas of high pollutant loads and potential management practices to reduce the loads. In-stream variables include stream discharge, sediment(t), total nitrogen (kg), and total phosphorus (kg). These variables are compared with observation data and are used to calibrate and validate the SWAT model. In-stream variables are important because they represent the impact of land use and management practices on water quality and aquatic ecosystems.

Calibration and validation of the SWAT model involve adjusting model parameters to improve the model's fit with observation data for in-stream variables. Once calibrated, the model can be used to evaluate the impacts of different land management scenarios on water quality and ecosystem health.

Model calibration was conducted in three watersheds using SWAT model options with SWAT-CUP [36]. The model outputs and USGS gauging data were calibrated using the Sequential Uncertainty Fitting Ver.2 (SUFI-2) in SWAT-CUP [36]. A sensitivity analysis of SWAT parameters was conducted before calibration. The most sensitive parameters were found to be SCS curve number for moisture condition II (CN2), available water capacity of the soil layer (SOL_AWC), USLE equation support practice factor (USLE_P), soil erodibility factor (USLE_K), and average slope steepness (HRU_SLP). The selected parameters were chosen based on the literature [23,37] and on former modeling research about these watersheds [14,38–40]. One-at-a-time local sensitivity and global sensitivity analysis were undertaken to select the SWAT model's most sensitive parameters [41,42]. All three SWAT models were calibrated with the desired sensitive parameters on the available time-series data: hydrology, sediment, total nitrogen, and total phosphorus [43]. For the Greensboro watershed, calibration was conducted between 1993 and 1999, and validation was between 2000 and 2002 on a daily time step. For the Watts Branch watershed, the model runoff was calibrated between 2003 and 2009 and validated from 2010 through 2012. Sediment, total nitrogen, and total phosphorus were calibrated between 2008 and 2009 as limited measured data were only available for these two years. In Watershed 263, only

nitrogen and phosphorus data from 2004 through 2010 were available, so total nitrogen and phosphorus were calibrated between 2004 and 2008 and validated from 2009 to 2010. After calibration and validation, SWAT models for these three watersheds were rerun with the new parameters for ten years. The watershed responses were analyzed based on the calibrated model output.

To evaluate the performance of model calibration and validation, we choose coefficient of determination ($R^2$), Nash-Sutcliffe model efficiency coefficient (*NSE*), and percentage of bias (*PBIAS*) as the evaluation criteria. Correlation of determination examines the linear relationship between observations and model outputs. *NSE* is a normalized statistic that determines the relative magnitude of the residual variance compared to the measured data variance and assesses the degree a model explains the observations. *PBIAS* assesses the fitness level between simulation and observations [44]. Calculations of $R^2$, *NSE*, and *PBIAS* are shown below:

$$R^2 = \frac{\left[ \sum_i \left( Q_{m,i} - Q_m \right) \left( Q_{s,i} - Q_s \right) \right]^2}{\sum_i \left( Q_{m,i} - \bar{Q}_m \right)^2 \sum_i \left( Q_{s,i} - \bar{Q}_s \right)^2} \tag{1}$$

$$NSE = 1 - \frac{\sum_i (Q_{m,i} - Q_{s,i})^2}{\sum_i \left( Q_{m,i} - Q_m \right)^2} \tag{2}$$

$$PBIAS = \frac{\sum_i (Q_{s,i} - Q_{m,i})}{\sum_i Q_{m,i}} \times 100 \tag{3}$$

where

$Q_{m,i}$—observed values at time step *i*;

$Q_m$—average observed value;

$Q_{s,i}$—simulated value at time step *i*;

$\bar{Q}_s$—average simulated value.

The coefficient of determination $R^2$ is between 0 and 1. 0 means no relationship, while 1 is for perfect correlation. Nash-Sutcliffe coefficient *NSE* is between $-\infty$ and 1. If *NSE* is negative, the average value of the measured value is better than the model simulation output. *PBIAS* measures the average tendency of the simulated data to be larger or smaller than the observations. A proper *PBIAS* should be smaller than 55% [44]. These three parameters are used as the objective function for model calibration and validation in SWAT-CUP.

### 2.3. Synthetic Watershed Model Development

As synthetic watersheds were simulated in a controlled and well-defined environment, they can provide valuable information of the impacts of different land covers on NPS pollutants output. The land covers of the Watts Branch watershed were changed to build three synthetic watersheds that allowed us to compare the effects of land cover while controlling for the effects of soil, slope, and weather. The details of the three synthetic watersheds are summarized in Figure 3 and Table 2. The original Watts Branch watershed (Figure 3c Baseline/B) is a suburban watershed. About 60% of the watershed area is in low-density or medium-density residential areas, 12% is covered with forest, and the remaining areas are high-density residential and industrial areas. To achieve the synthetic natural watershed, the low-density and medium-density residential subareas were changed to forest; the high-density and industrial areas were converted to low-density and medium-density areas in order to build the synthetic natural watershed. This natural synthetic watershed is established by having most of the areas in forest after the conversion. Hence, the first synthetic natural watershed (Figure 3a, D1) has more than 72% natural (i.e., forest) areas. For the synthetic agricultural watershed (Figure 3b, D2), the baseline watershed's low-density, medium-density, and high-density residential areas were all changed to agricultural

areas, which accounted for more than 75% of the total area. For the synthetic urban watershed (Figure 3d, D3), the baseline watershed's low-density, medium-density, and high-density residential areas were changed to the industrial area, which could be defined as an ultra-urban watershed. The soil, slope, and weather data were unchanged for these three synthetic watersheds. Three SWAT model input parameter options were built based on these settings, and the HRU outputs were retrieved for further analysis. Calibration and validation were not needed in the synthetic watersheds. As we only concentrate on the spatial distribution of NPS pollutants instead of the exact NPS output concentration, watersheds that lack data for calibration are still helpful for understanding the impact of land use on water quality.

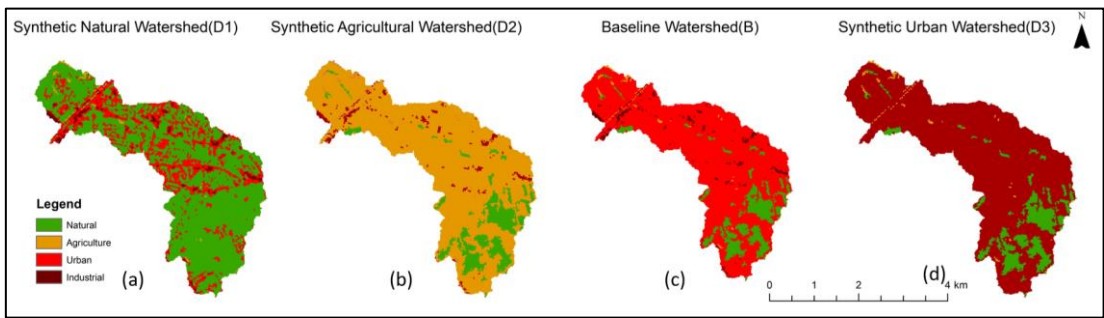

**Figure 3.** Land cover for three synthetic watersheds from the original baseline watershed: (**a**) synthetic natural watershed (D1); (**b**) synthetic agricultural watershed (D2); (**c**) original Watts Branch watershed (B) as the baseline watershed; (**d**) synthetic urban watershed (D3).

**Table 2.** Land cover for three synthetic watersheds.

| | Synthetic Natural Watershed (D1) | | Synthetic Agricultural Watershed (D2) | | Baseline Watershed (B) | | Synthetic Urban Watershed (D3) | |
|---|---|---|---|---|---|---|---|---|
| | Area (ha) | Percentage (%) | Area (ha) | Percentage (%) | Area (ha) | Percentage (%) | Area (ha) | Percentage (%) |
| Natural | 751.35 | 72.15 | 134.26 | 12.90 | 134.26 | 12.90 | 134.26 | 12.90 |
| Agriculture | 3.66 | 0.35 | 871.98 | 83.77 | 3.66 | 0.35 | 3.66 | 0.35 |
| Urban Residential | 286.25 | 27.5 | 0 | 0 | 868.32 | 83.42 | 0 | 0 |
| Industrial | 0 | 0 | 34.66 | 3.33 | 34.66 | 3.33 | 902.98 | 86.85 |

## 3. Results and Discussions

### 3.1. SWAT Model Performances

The performance trends and statistics of the calibrated and validated SWAT model for each watershed are shown in Figure 4 and Table 3, respectively.

The Greensboro watershed was divided into 21 subbasins with 7780 HRUs. SWAT model's calibration and validation, as was done by Renkenberger et al. (2016) [38], was done with SurfQ, Sed, Tot N, and Tot P between 1993 and 2002. Calibration of total flow was first conducted on a daily time step. Then calibrations of Sed, Tot N, and Tot P were performed in sequence. The correlation coefficient values are suitable for all four outputs of interest (i.e., SurfQ, Sed, Tot N, and Tot P). The $R^2$ values for all of them were higher than 0.60 on an annual basis, indicating a strong linear correlation between the measured and simulated values. As for *NSE*, which means how well the plot of observed versus simulated data fits the line of equal values (i.e., 1:1 line), was suitable for SurfQ (0.51), Sed (0.57), Tot P (0.79), and acceptable for Tot N (0.47) [44]. The percentage of sediment bias was 50.12%, which meant that the sediment output by SWAT model was overestimated. Based on the given statistics, we considered the SurfQ, Tot N, and Tot P predictions to be underestimated, but all values were within the acceptable ranges [44,45].

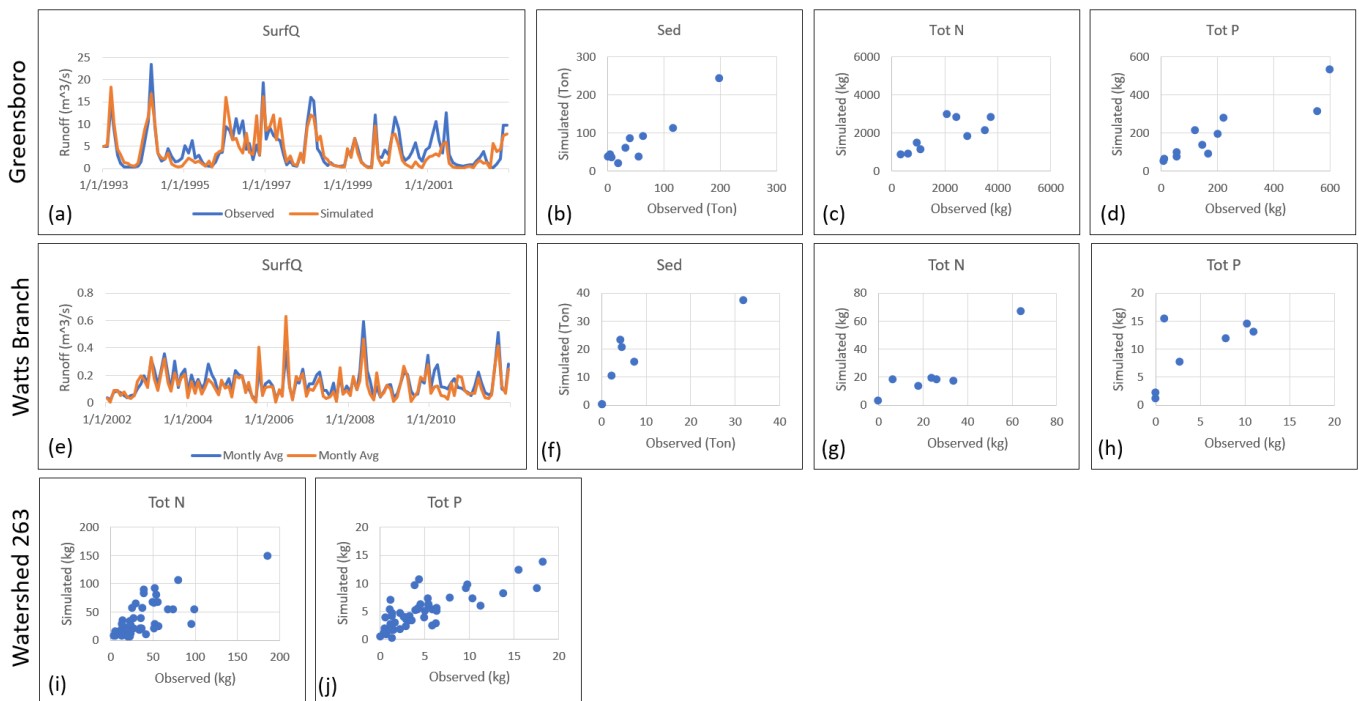

**Figure 4.** Performance trends of the SWAT calibration and validation in three watersheds: (**a–d**) Greensboro, (**e–h**) Watts Branch, and (**i,j**) Watershed 263.

**Table 3.** Performance statistics of SWAT model for the study watersheds in simulating total flow (SurfQ), total sediment (Sed), total nitrogen (Tot N), and total phosphorus (Tot P), 1993 to 2002 in Greensboro, and 2002 to 2011 in Watts Branch and Watershed 263.

| | Greensboro Watershed (Annual) | | | Watts Branch Watershed (Daily) | | | Watershed 263 (Annual) | | |
|---|---|---|---|---|---|---|---|---|---|
| | $R^2$ | NSE | PBIAS (%) | $R^2$ | NSE | PBIAS (%) | $R^2$ | NSE | PBIAS (%) |
| SurfQ | 0.64 | 0.51 | −10.17 | 0.76 | 0.73 | 17.4 | | | |
| Sed | 0.85 | 0.57 | 50.12 | 0.75 | 0.57 | 57 | | NON | |
| Tot N | 0.60 | 0.47 | −13.2 | 0.77 | 0.79 | −9.2 | 0.67 | 0.50 | 5.2 |
| Tot P | 0.83 | 0.79 | −5.83 | 0.92 | 0.45 | 58.2 | 0.66 | 0.50 | 5.3 |

The Watts Branch was divided into 22 subbasins with 2148 HRUs. Calibration and validation of the SWAT model were conducted on a daily time step. The model's runoff simulations were first calibrated using the available data between 2002 and 2008 and validated between 2009 and 2011 on a daily time step. As the Watts Branch watershed has limited measured nutrient data in 2008 and 2009, the sediment, total nitrogen, and total phosphorus were calibrated for the period from 2008 through 2009. The performance statistics of the calibrated and validated model were shown in Table 3. The $R^2$ values were excellent for all four outputs. *NSE* and *PBIAS* values were good, except *NSE* was 0.45 for total phosphorus, and *PBIAS* was 58.2% for total phosphorus and 57% for total sediment. Poor performance on sediment simulations translated to the poor performance of phosphorus, considering that most of the phosphorus leaved the watershed as sediment-bound.

Watershed 263 was divided into 23 subbasins with 1006 HRUs. Since only nitrogen and phosphorus were available in this watershed, the SWAT model was calibrated only based on total nitrogen and total phosphorus data during 2004 and 2008 and validated during 2009 and 2010, all on a monthly time step. Statistics of Watershed 263 showed $R^2$, *NSE*, and *PBIAS* were 0.67, 0.5, and 5.2%, for Tot N, respectively, and the results for the total phosphorus were very similar, with values of $R^2$, *NSE*, and *PBIAS* being 0.66, 0.5, and

5.3%, respectively. This indicated SWAT's consistent performance for both nitrogen and phosphorus simulations in watershed 263.

Synthetic natural, agricultural, and urban watersheds were delineated into 1293, 1118, and 1006 HRUs, respectively, and SWAT simulations were run in ten years as the other three real watersheds. Models' output of selected constituents are presented in the following sections.

### 3.2. Spatial Distribution of NPS Constituents in the Three Study Watersheds

3.2.1. Mass-Area Ratio for Three Watersheds

Mass-area ratios (i.e., the percentage of each constituent yield versus the total area) are shown in Figure 5. The portion of the total yields is much higher than the percentage of total areas that produce these pollutant yields in all three real watersheds (i.e., the hotspots), but the trend of the curve lines for each constituent in each watershed is different. Sed is the most concentrated whereas SurfQ is the least concentrated, across all watersheds. For the agricultural Greensboro watershed (Figure 5a), the accumulated mass-area lines of Sed, Tot N, and Tot P are similar and all higher than the SurfQ lines, meaning that their respective pollution hotspots are more concentrated than for SurfQ. When we consider the top 20% of HRUs by exported pollution concentration as hotspots for each pollutant, total areas account for ~50% output for Sed, Tot N, and Tot P but only 35% for SurfQ (Figure 5a). The suburban watershed (Watts Branch) is shown in Figure 5b. The accumulated mass-area ratio map shows that Sed, Tot P, Tot N, and SurfQ are similarly concentrated in hotspots in this watershed, but Sed and Tot P are more concentrated than Tot N and SurfQ. For the ultra-urban Watershed 263 (Figure 5c), the mass-area curve lines for Sed and Tot P are much more curved than for SurfQ and Tot N, indicating that sediment and phosphorus discharges are more concentrated hotspots in the watershed. SurfQ and Tot N curves for this watershed are near the 1:1 line, meaning that these pollutants are relatively evenly spatially distributed across the landscape. In contrast, Sed and Tot P are more concentrated into hotspots. Mass-area Sed and Tot P lines are similar in three watersheds, inferring that phosphorus mainly discharges as sediment-bounded phosphorus.

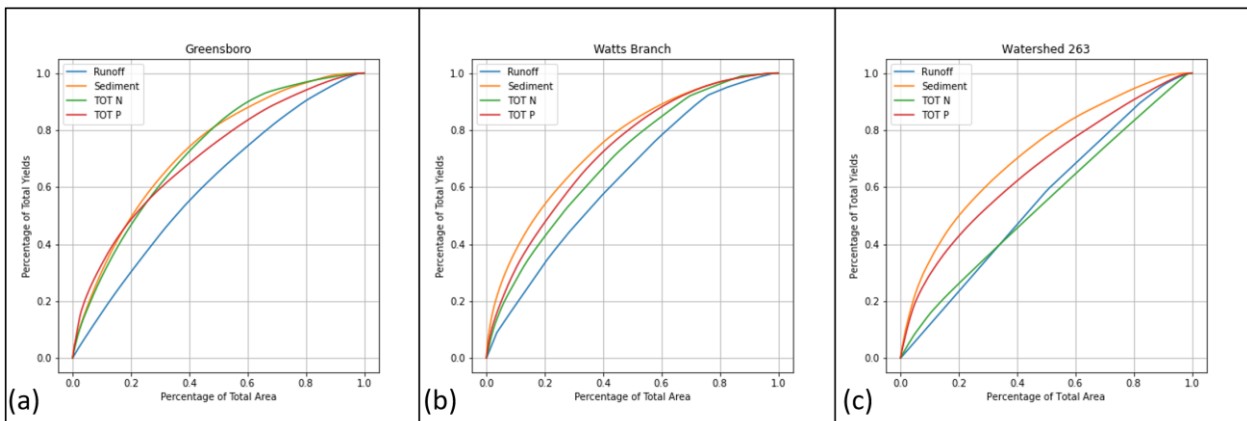

**Figure 5.** Mass-area ratios of total surface runoff (SurfQ), total sediment (Sed), total nitrogen (Tot N), and total phosphorus (Tot P) in each real watershed: (**a**) Greensboro watershed (agricultural), (**b**) Watts Branch watershed (suburban), (**c**) Watershed 263 (ultra-urban).

3.2.2. Comparison of the Same Pollutant among Different Watersheds

The comparations of each pollutant output in three real watersheds (i.e., Greensboro, Watts Branch, and Watershed 263) are shown in Figure 6. In Figure 6a, all surface runoff curves are near the 1:1 line, and the runoff yields for all three watersheds are approximately similar. This indicates that SurfQ hotspots tend to be evenly distributed in all three watersheds. However, it should be noted that runoff origination is still more concentrated in certain hotspots in suburban (Watts Branch) and agricultural (Greensboro) watersheds.

Figure 6b shows that the mass-area curve lines of Sed are similar in these three watersheds, thus indicating sediment hotspots' spatial distributions are similar. According to Figure 6c, the concentration of Tot N hotspots in the Greensboro watershed and Watts Branch watershed are more significant than in Watershed 263. Moreover, it should be noted that mass-area yield for Tot N in the Greensboro watershed is higher than in Watts Branch Watershed despite more stakeholder engagement in BMP implementation in the Greensboro watershed. It may also be due to the fact that for the same percent area in the large Greensboro watershed, a larger area is contributing to Tot N discharge than the Watts Branch watershed. Tot N is evenly distributed in Watershed 263 (Figure 6c). In combination with Figures 5c and 6c, nitrogen in surface flow accounted for the most total nitrogen output in the ultra-urban watershed. This percentage decreases more and more in suburban and agricultural watersheds. Similar to SurfQ and Sed, Tot P concentration is the least spatially concentrated in Watershed 263 (Figure 6d). In summary, all four constituents' outputs are the least spatially concentrated in watershed 263; the SurfQ, Tot N, and Tot P areal concentrations' differences are more significant than the Sed areal concentration in all three watersheds.

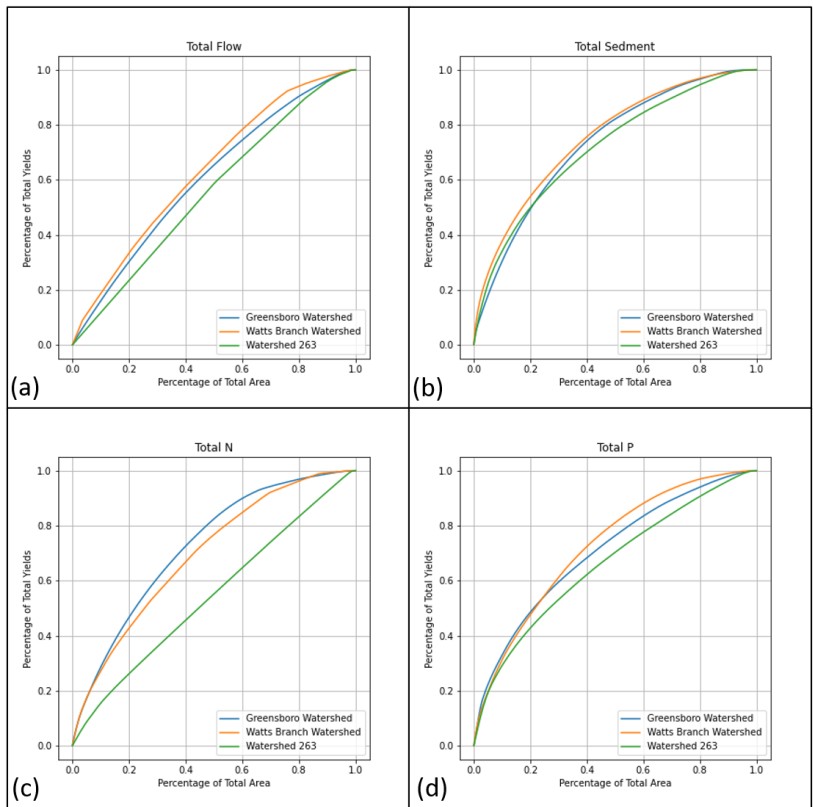

**Figure 6.** Mass-area ratios of (**a**) total surface flow (SurfQ), (**b**) total sediment (Sed), (**c**) total nitrogen (Total N), and (**d**) total phosphorus (Total P) among three real watersheds.

### 3.2.3. Hotspots' Spatial Distribution

In general, an area is considered a hotspot if the total constituent output percentage is much higher than the percentage of the area. The hotspots are identified by (1) calculating the output of each HRU and the total output of the whole watershed; (2) calculating the percentage output of each HRU; (3) ranking all HRUs by their density of SurfQ (mm), Sed (Ton/ha), Tot N (kg/ha), Tot P (kg/ha); (4) accumulating sum of constituents in HRUs based on the rank until reaching 20% and 50% of the total output of each constituent, marking these HRUs as hotspots of SurfQ, Sed, Tot N, and Tot P. In the three real watersheds, hotspots are responsible for the top 20% and top 50% of total constituent outputs. The chosen hotspot thresholds are shown in Table 4, and the spatial distribution is in Figure 7. For example, the chosen hotspots for SurfQ in Greensboro watershed are the HRUs with

SurfQ output density higher than 345.63 mm, which account for 20% of total surface runoff output by just 12.63% of total areas. As for sediment, the HRUs with sediment output density higher than 7.7 t/ha are chosen as sediment hotspots in Watts Branch, which only cover 3.09% of total areas but produce 20% of total sediment output.

**Table 4.** Hotspot thresholds in three real watersheds.

| | Output (%) | SurfQ (mm) | | Sed (t/ha) | | Tot N (kg/ha) | | Tot P (kg/ha) | |
|---|---|---|---|---|---|---|---|---|---|
| | | 20 | 50 | 20 | 50 | 20 | 50 | 20 | 50 |
| Greensboro watershed | area (%) | 12.63 | 33.48 | 5.7 | 20.35 | 6.2 | 22.08 | 4.08 | 21.13 |
| | Threshold | 345.63 | 273.84 | 4.146 | 1.54 | 13.02 | 10.79 | 5. 60 | 2.58 |
| Watts Branch | area (%) | 10.91 | 33.29 | 3.09 | 17.19 | 6.2 | 25.45 | 5.11 | 21.7 |
| | Threshold | 358.01 | 241.14 | 7.70 | 4.99 | 23.51 | 21.63 | 6.81 | 4.40 |
| Watershed 263 | area (%) | 17.1 | 42.65 | 4.31 | 20.09 | 14.05 | 44.66 | 5.21 | 26.72 |
| | Threshold | 811.51 | 811.14 | 8.44 | 3.53 | 19.11 | 17.98 | 6.49 | 3.49 |

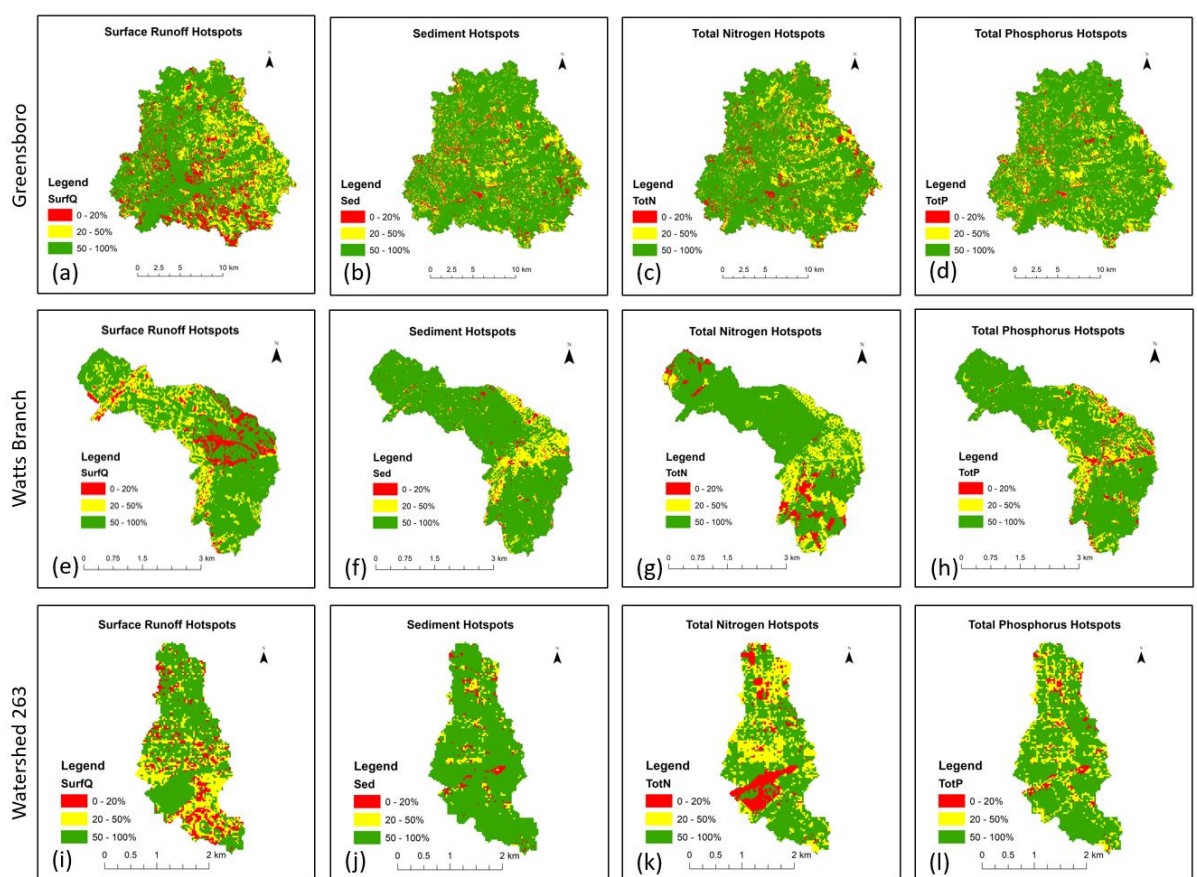

**Figure 7.** Hotspots of (**a**,**e**,**i**) total surface runoff (SurfQ), (**b**,**f**,**j**) total sediment (Sed), (**c**,**g**,**k**) total nitrogen (Tot N), and (**d**,**h**,**l**) total phosphorus (Tot P) in three real watersheds (first row: Greensboro, second row: Watts Branch, third row: Watershed 263). Red and yellow areas in each map account for 20% and 30% of the total output of each constituent, respectively.

About 12% of the watershed area accounts for 20% of SurfQ in Greensboro Watershed and Watts Branch Watershed, whereas 17% of the region accounts for 20% of SurfQ in Watershed 263. SurfQ is more concentrated in certain areas in agricultural and suburban watersheds than in ultra-urban watersheds where it is spatially spread out. Comparing the SurfQ map in Figure 7a,e,i and the land cover map (Figure 2), the SurfQ hotspots

are not located in the natural areas, rather they are in the urban or agricultural areas as expected. Sediment hotpots are the most concentrated among the four target constituents. Sediment hotspots are in agricultural, low-density, and medium-density residential areas with relatively high slopes. The Tot N output distribution differs between Watershed 263 and the other two watersheds. In Watershed 263, 14% of the watershed hotspot areas account for 20% of Tot N output, while in the Greensboro and Watts Branch, the hotspot area is about 6%. Therefore, Tot N discharge areas are more concentrated in specific areas in agricultural and suburban watersheds than in ultra-urban watersheds, where they are distributed to more spots. The distribution of nitrogen hotspots shows that their hotspots are related to hydrologic soil group C, where the infiltration rate is higher than the soils in hydrologic group D but lower than the hydrologic groups A and B. Compared the red areas in sediment and phosphorus hotspots in three watersheds, the locations of phosphorus hotspots are similar to sediment hotspots in three watersheds; this may be due to the fact that phosphorus is often attached to sediment and flow out by surface flow.

*3.3. Confirming Distribution Pattern in Synthetic Watersheds*

3.3.1. Mass-Area Ratio in Three Synthetic Watersheds

The mass-area ratio for three synthetic watersheds is shown in Figure 8. For the synthetic natural watershed (D1) in Figure 8a, Sed and Tot N are much more concentrated in a small percentage of watershed areas than SurfQ and Tot P. 20% of the hotspot area accounts for about 80% of Sed and Tot N yield, 45% of Tot P output, and 35% of SurfQ. For synthetic agricultural watershed (D2) (Figure 8b), the accumulated trends of nitrogen and phosphorus outputs are similar. At the same time, they are lower than the areal concentration of sediment and more concentrated than surface runoff. 20% of top hotspot areas account for 70% of sediment, 50% of Tot N and Tot P, and 35% of SurfQ. Runoff and nitrogen are evenly distributed for the synthetic ultra-urban watershed (D3), as shown in Figure 8c, while sediment and phosphorus are concentrated in a small percentage of the watershed area. 20% of top hotspots produce 45% of sediment, 35% of Tot P, and 25% of Tot N and surface runoff.

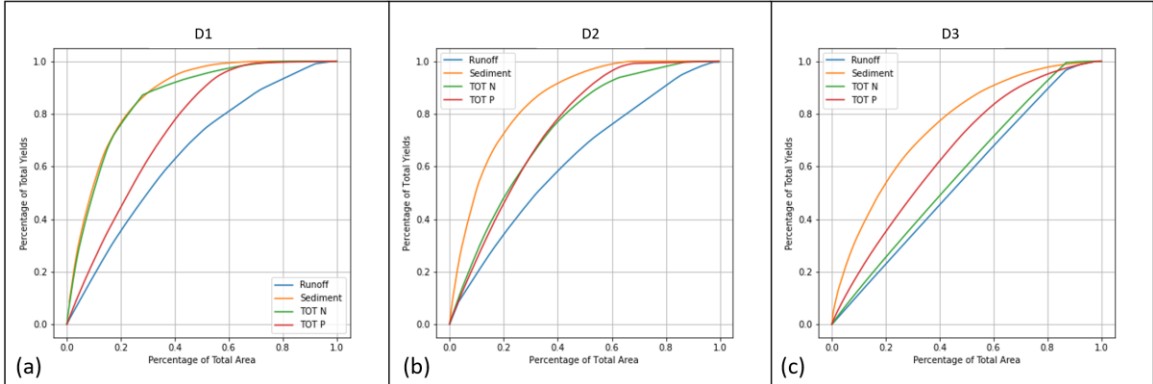

**Figure 8.** Mass-area ratios of surface runoff (SurfQ), total sediment (Sed), total nitrogen (Tot N), and total phosphorus (Tot P) in three synthetic watersheds: (**a**) synthetic natural watershed (D1), (**b**) synthetic agricultural watershed (D2), (**c**) synthetic urban watershed (D3).

In conclusion, sediment hotspots are the most concentrated, while runoff is the least concentrated in these watersheds, and patterns could also be found in the real watersheds. As the land cover changes from forest to agricultural and urban areas, the outputs of surface runoff, sediment, Tot N, and Tot P become more and more evenly distributed.

3.3.2. Comparison of NPS Pollutants among Synthetic Watersheds

Comparison of the four constituents (SurfQ, Sed, Tot N, and Tot P) in three synthetic watersheds are summarized in Figure 9. SurfQ in synthetic natural (D1) and synthetic agricultural (D2) watersheds are similar, and they are more spatially concentrated than the

synthetic urban watershed (D3), where they are spatially and evenly distributed. Curve lines of Sed are similar in the synthetic natural watershed (D1) and synthetic agricultural watershed (D2), and they are much more spatially concentrated than in the synthetic urban watershed (D3). Tot N output is the most spatially concentrated in the synthetic natural watershed (D1), followed by the synthetic agricultural watershed (D2) and synthetic urban watershed (D3). Tot P load concentration is similar in the synthetic natural watershed (D1) and synthetic agricultural watershed (D2), and they are more spatially concentrated than the synthetic urban watersheds (D3). In conclusion, as land cover changes from natural to urban areas, SurfQ, Sed, Tot N, and Tot P become more evenly distributed over the watershed area.

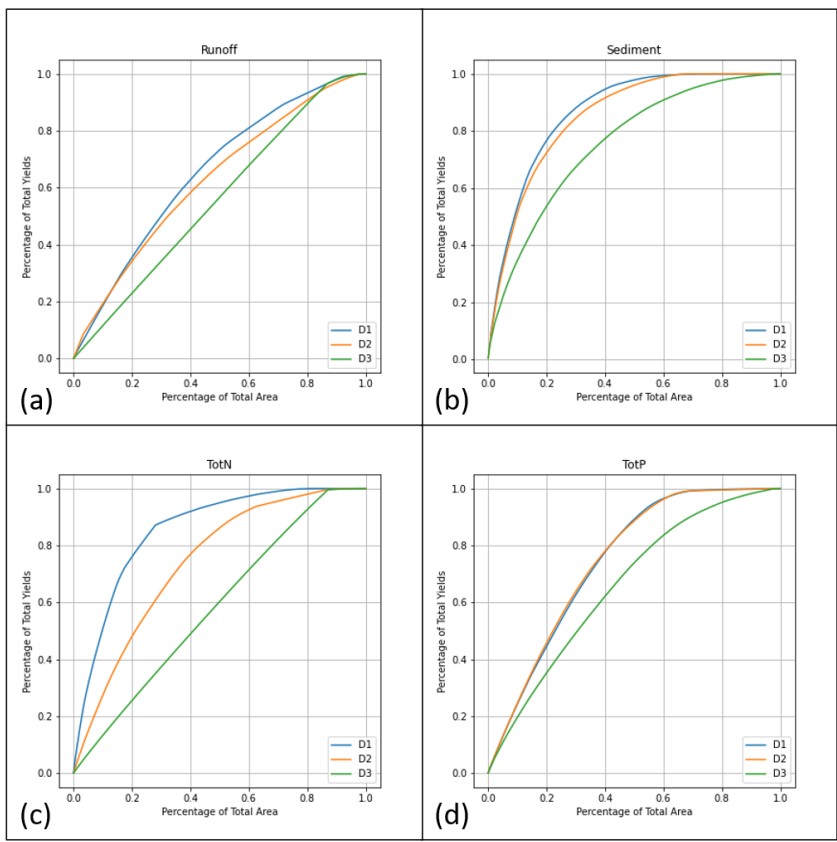

**Figure 9.** Mass-area ratios of (**a**) surface runoff (SurfQ), (**b**) total sediment (Sed), (**c**) total nitrogen (Tot N), and (**d**) total phosphorus (Tot P) among three synthetic watersheds: (D1) synthetic natural watershed, (D2) synthetic agricultural watershed, (D3) synthetic urban watershed.

### 3.3.3. Spatial Distribution of Hotspots in Synthetic Watersheds

The thresholds that were chosen for constituents' hotspots are shown in Table 5, and the spatial locations of these hotspots are shown in Figure 10. Most of the hotspots are in the areas of the watershed with high slopes and low infiltration. SurfQ is more concentrated in certain areas in the synthetic natural (D1) and synthetic agricultural (D2) watersheds than in the synthetic urban watershed (D3). The top 20% SurfQ hotspots are in the watershed areas with fine-textured soils belonging to the hydrologic soil group D where the infiltration rate is the lowest in the whole watershed. As such, most water does not infiltrate and runs off as surface runoff. Sed hotspots are highly concentrated in three synthetic watersheds. For example, 20% of total sediment output is due to 2.7% of total land hotspots in the synthetic natural (D1) and synthetic agricultural (D2) watersheds and 4.5% of the total area hotspots in the synthetic urban watershed (D3). Most sediment hotspots are in the watershed areas with hydrologic soil groups C and D and slopes higher than 5%. For Tot N, the top 20% output, as red color in Figure 10c,g,k, is becoming more and more evenly distributed from the synthetic natural watershed (D1) to the synthetic urban watershed (D3). In synthetic

urban watershed (D3), the nitrogen hotspots are located on the edges between urban and natural areas. Tot P is much more concentrated in the synthetic urban watershed (D3) than in the synthetic natural watershed (D1) and the synthetic agricultural watershed (D2).

**Table 5.** Hotspots thresholds in three synthetic watersheds.

| | | SurfQ (mm) | | Sed (t/ha) | | Tot N (kg/ha) | | Tot P (kg/ha) | |
|---|---|---|---|---|---|---|---|---|---|
| | Output (%) | 20 | 50 | 20 | 50 | 20 | 50 | 20 | 50 |
| D1 | Area (%) | 10.81 | 29.83 | 2.71 | 9.05 | 2.84 | 9.78 | 8.12 | 22.83 |
| | Threshold | 307.70 | 242.42 | 2.636 | 1.404 | 8.21 | 4.93 | 5.90 | 4.84 |
| D2 | Area (%) | 10.65 | 32.48 | 2.74 | 9.47 | 6.98 | 21.19 | 7.96 | 22.40 |
| | Threshold | 420.68 | 307.99 | 6.273 | 3.778 | 25.77 | 16.97 | 10.31 | 8.34 |
| D3 | Area (%) | 17.29 | 44.09 | 4.45 | 18.03 | 15.46 | 41.00 | 9.87 | 30.67 |
| | Threshold | 688.60 | 683.26 | 9.513 | 5.140 | 11.94 | 11.24 | 11.09 | 8.78 |

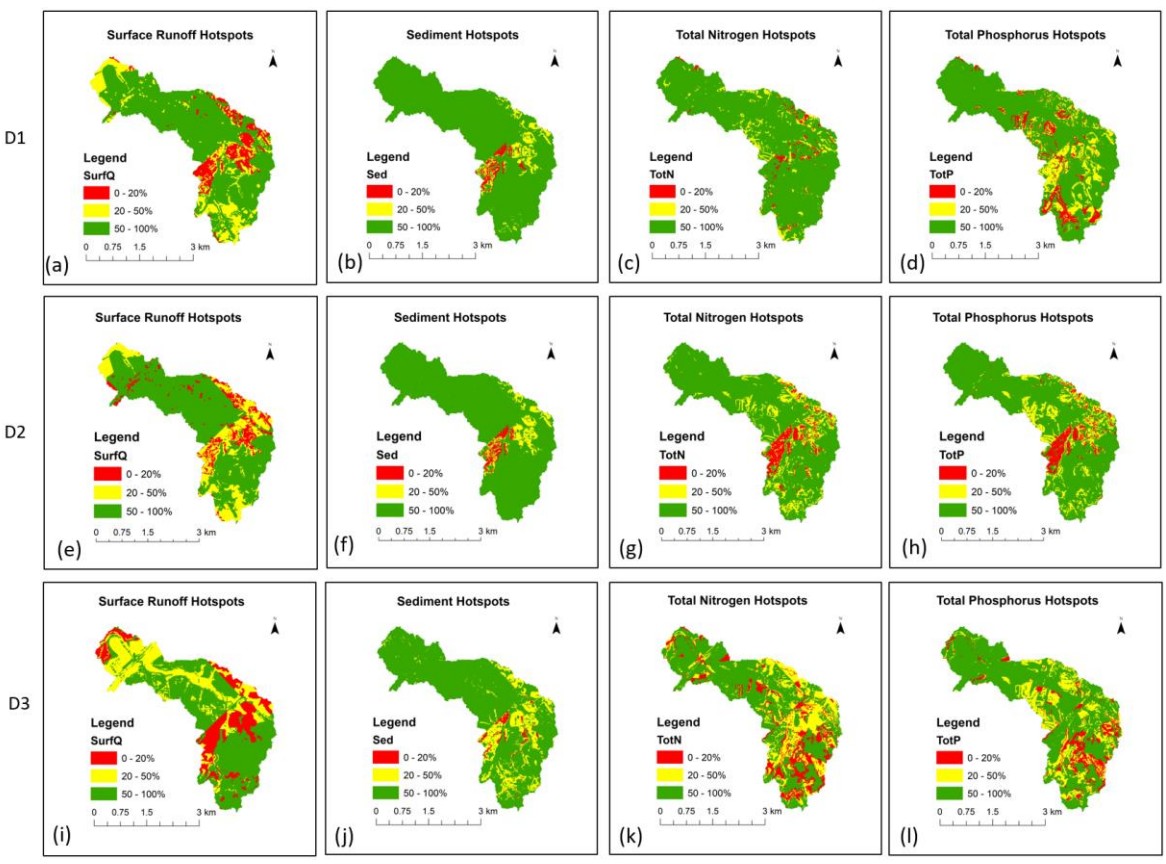

**Figure 10.** Hotspots for (**a**,**e**,**i**) total surface runoff (SurfQ), (**b**,**f**,**j**) total sediment (Sed), (**c**,**g**,**k**) total nitrogen (Tot N), and (**d**,**h**,**l**) total phosphorus (Tot P) in three synthetic watersheds: (D1) synthetic natural watershed, (D2) synthetic agricultural watershed, and (D3) synthetic urban watershed. Red and yellow areas in each map account for 20% and 30% of the total output of each constituent, respectively.

## 4. Conclusions

This study aimed to investigate the influence of land cover on the concentration of surface runoff and three important pollutants including total sediment, total nitrogen, and total phosphorus in three real and three synthetic watersheds in the Chesapeake Bay basin. The real watersheds represented agricultural, suburban, and ultra-urban areas, while the synthetic watersheds represented natural, agricultural, and urban regions. The choice of watersheds was based on the fact that land cover varies as communities develop, ranging from natural to suburban and ultra-urban. SWAT modeling framework was used

to simulate the hydrologic process in the study watersheds, and model outputs were analyzed accordingly.

The results indicated that pollutant hotspots became increasingly uniform as land cover changed from natural to urban, with sediment being the most concentrated pollutant. In natural or agricultural areas where pollutant hotspots are more concentrated, the study suggests that managing NPS pollution is easier due to the smaller area of the watershed requiring the installation of best management practices and coordination with fewer landowners. However, in urban areas, particularly ultra-urban regions, the study suggests that controlling NPS pollution hotspots requires more widespread strategies and coordination with more stakeholders.

Overall, the findings of this study have significant implications for NPS pollution management, highlighting the importance of developing effective strategies for managing NPS pollution, particularly sediment and nutrients, in both rural and urban areas as communities develop and land cover changes. Policymakers and land managers can use the results of this study to develop targeted and efficient management strategies that are tailored to the specific land cover types and the severity of NPS pollution to ensure the protection of water quality and the sustainability of ecosystems.

As for the limitations of the study, firstly, the research findings are based on the assumption that the simulation framework SWAT can accurately represent the hydrological processes of the study watersheds. Although the models have been calibrated and validated, it is important to confirm the findings of this research with other modeling methods and field survey studies. Secondly, the incomplete daily gauging data for all three watersheds could introduce biases and affect the model performance. Therefore, it is essential to obtain complete and accurate data for future studies. Lastly, this research only focuses on three watersheds in the Chesapeake Bay Basin in the U.S. Hence, it is necessary to evaluate the findings in other areas of the U.S. or the world to determine the generalizability of the study results.

In the future, it is recommended to conduct research in other areas to test the generalizability of the current findings. Additionally, it is important to incorporate land cover information into existing Best Management Practice (BMP) allocation tools to account for the spatial distribution of NPS constituents across different land cover types. In urban areas where NPS hotspots are more spatially distributed, investigating BMP adoption behavior and identifying effective social interventions for NPS pollution control are crucial. Furthermore, quantifying the social impacts of BMP adoption can inform effective social interventions. These recommendations aim to enhance our understanding of NPS pollution control and guide the development of more effective BMP allocation strategies.

**Author Contributions:** Conceptualization, Z.Z., H.M. and A.S.; methodology, Z.Z. and H.M.; software, Z.Z.; validation, Z.Z., H.M. and A.S.; formal analysis, Z.Z.; investigation, Z.Z.; resources, H.M., A.S. and P.T.L.; data curation, Z.Z.; writing—original draft preparation, Z.Z. and H.M.; writing—review and editing, H.M., A.S., P.T.L. and M.N.-A.; visualization, Z.Z.; supervision, H.M. and A.S.; project administration, P.T.L.; funding acquisition, H.M., A.S., M.N.-A. and P.T.L. All authors have read and agreed to the published version of the manuscript.

**Funding:** This research was funded by the National Science Foundation-Coupled Natural Human Systems award (DEB 1824807).

**Data Availability Statement:** Not applicable.

**Conflicts of Interest:** The authors declare no conflict of interest.

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
