# Peer review of "Impacts of Land Cover Change on the Spatial Distribution of Nonpoint Source Pollution Based on SWAT Model"

_water, doi:10.3390/w15061174_

Round 1

Reviewer 1 Report

The authors of the manuscript entitled “Impacts of land cover change on the spatial distribution of non-point source pollution based on SWAT model” aims at evaluating the spatial distribution of NPS hotspots under the scenario of land cover change. This study is interesting for the WATER community, because it is focusing on NPS pollutant output and guide control measures in urbanized basins. Overall, the manuscript is well-logical and easy to comprehend with a good English language. I have some moderated concerns for the manuscript before publication.

1. The abstract is tedious, need to be refined.
2. The global sensitivity analysis allows the selection of the most sensitive parameters that affect the SWAT model. The sensitivity parameters selected in the present paper were chosen based on the literature 21 and 33. Which of parameters are the most sensitive in this study? Is it the same as in the literature 21 and 33? Which parameters did you choose when you performed the parameter calibration?

3. Please add a figure of the SWAT calibration and verification process.

4. I suggest using the coefficient of determination (R2) to evaluate the model, which is used by more related scholars and is one of the important indicators for SWAT model evaluation.

5. How the conversion of land use in the three synthetic watersheds was achieved, the detailed process of land use conversion is lacking in the text, please supplement this part.

6. Why is the Watts Branch watershed divided into 22 subbasins and what is the basis for this? As far as I know, the number of subbasins in SWAT model can be varied by setting the relevant threshold, what is the threshold you set? In addition, the division of HRU is based on the division of subbasins, and the number of subbasins changes by setting different thresholds, so will the number of HRU also change in this paper? Thus, will your settings influence the simulation results as well? Because the division of HRU is done within subbasins, no cross-basin HRU exist.

7. Please specify the basis of subbasin division.

8. “Results and Discussions”, need the authors to refine the results and discuss the results in depth.

9. Based on the findings, more useful measures for controlling NPS in urbanized basins should be recommended in the manuscript.

Author Response

  1. The abstract is tedious, need to be refined.

The abstract has been refined.

  1. The global sensitivity analysis allows the selection of the most sensitive parameters that affect the SWAT model. The sensitivity parameters selected in the present paper were chosen based on the literature 21 and 33. Which of parameters are the most sensitive in this study? Is it the same as in the literature 21 and 33? Which parameters did you choose when you performed the parameter calibration?

SCS curve number for moisture condition II (CN2) was the most sensitive parameter in the experiments. This is consistent with the literature. In addition to CN2, we also calibrated the model using other parameters such as SOL_AWC, USLE_P, USLE_K, and HRU_SLP.

  1. Please add a figure of the SWAT calibration and verification process.

A new figure (figure 4) showing the calibration and validation trends was added as figure 4 in the manuscript.

  1. I suggest using the coefficient of determination (R2) to evaluate the model, which is used by more related scholars and is one of the important indicators for SWAT model evaluation.

Yes, we have changed R2 instead of r to evaluate the model in the paper.

  1. How the conversion of land use in the three synthetic watersheds was achieved, the detailed process of land use conversion is lacking in the text, please supplement this part.

We have discussed the details of the land conversion process in section “2.3 Synthetic watershed model development”:

“The details of the three synthetic watersheds are summarized in Figure 3 and Table 2. The original Watts Branch watershed (Figure 3(c) Baseline/B) is a suburban watershed. About 60% of the watershed area is in low-density or medium-density residential areas, 12% is covered with forest, and the remaining areas are high-density residential and industrial areas. To achieve the synthetic natural watershed, the low-density and medium-density residential subareas were changed to forest, the high-density and industrial areas were converted to low-density and medium-density areas in order to build the synthetic natural watershed. This natural synthetic watershed is established by having most of the areas in forest after the conversion. Hence, the first synthetic natural watershed (Figure 3(a) D1) has more than 72% natural (i.e., forest) areas. For the synthetic agricultural watershed (Figure 3(b) D2), the baseline watershed’s low-density, medium-density, and high-density residential areas were all changed to agricultural areas, which accounted for more than 75% of the total area. For the synthetic urban watershed (Figure 3 (d) D3), the baseline watershed’s low-density, medium-density, and high-density residential areas were changed to the industrial area, which could be defined as an ultra-urban watershed. The soil, slope, and weather data were unchanged for these three synthetic watersheds.”

  1. Why is the Watts Branch watershed divided into 22 subbasins and what is the basis for this? As far as I know, the number of subbasins in SWAT model can be varied by setting the relevant threshold, what is the threshold you set? In addition, the division of HRU is based on the division of subbasins, and the number of subbasins changes by setting different thresholds, so will the number of HRU also change in this paper? Thus, will your settings influence the simulation results as well? Because the division of HRU is done within subbasins, no cross-basin HRU exist.

The Hydrologic Response Unit (HRU) serves as the fundamental calculation unit in SWAT and is established based on a specific combination of land use, soil type, and slope class. In order to define HRUs for a particular subbasin when preparing the input file for SWAT, it is necessary to establish the thresholds as a percentage of the subbasin area for land use, soil type, and slope class. For instance, if a 5% threshold is set for a given land use within a particular subbasin, any land use that comprises less than 5% of the subbasin will not be associated with a new HRU. In other words, each HRU is a specific hydrologic unit with homogeneous land use, soil, and slope class.

To define HRUs for land use, soil, and DEM data, the number of HRUs increases as the threshold decreases. According to Yan Wang et al. (2016), decreasing the threshold improves the computational accuracy of the SWAT model. However, this also results in longer computational time. In this work, we selected 0% for land use, soil, and slope thresholds to achieve better computational accuracy.

  1. Please specify the basis of subbasin division.

The subbasin was automatically built with the stream orders, based on the topographic map of the given watershed defined in DEM (digital elevation model).

  1. “Results and Discussions”, need the authors to refine the results and discuss the results in depth.

We feel the results and discussions section is already described in depth. However, we made appropriate edits to make the section clearer to the readers. We edited calibration and validation figure 4 and made the model application clear. We also changed the correlation coefficient r to coefficient of determination R2 and discussed accordingly.  Figure 7 and figure 10 were also changed to make the watershed labeling clearer.

  1. Based on the findings, more useful measures for controlling NPS in urbanized basins should be recommended in the manuscript.

Thank you for this comment. Respectfully, the focus of this manuscript was to use SWAT model to identify critical NPS pollution across diverse watersheds and did not intend to prescribe control measures (i.e., BMPs such as green infrastructure) in pollution infected areas. This focus on control measures is being conducted under a different research program and will soon be reported to the literature as we make progress.

Reviewer 2 Report

this is an ordinary & solid modeling work (NPS study) based on SWAT. the modeling task has been validated by the authors, and I think the findings can substantially reflect the confident output of the designated experiment.

1. However, I think the results summarized in this paper should be more specific. I cannot get detailed information from their conclusions on how the LUCC disturb the NPS and how can we control pollution based on their findings. The results just tell the readers some literal common-sense findings, especially they did comparison analysis between different watersheds.

2. how to easily understand the curve figures such as 4, 5, or 7.

3. the hotpot results lack numerical results supportance and where are the detailed comparison between the baseline and designated watershed?

Author Response

  1. However, I think the results summarized in this paper should be more specific. I cannot get detailed information from their conclusions on how the LUCC disturb the NPS and how can we control pollution based on their findings. The results just tell the readers some literal common-sense findings, especially they did comparison analysis between different watersheds.

This study aimed to analyze the spatial distribution patterns of NPS pollutants and identify highly polluted areas or hotspots. The analysis was conducted by comparing the distribution of NPS constituents in three real watersheds and three synthetic watersheds. The results indicate that the distribution of pollutant hotspots becomes increasingly uniform as land cover changes from natural to urban areas. In natural or agricultural areas, pollutant hotspots are more concentrated, making managing NPS pollution easier due to the smaller area of the watershed requiring best management practices and coordination with fewer landowners. However, in urban areas, especially ultra-urban regions, controlling nonpoint source pollution hotspots requires more widespread strategies and coordination with more stakeholders more challenging. Results like this indicate different strategies that the responsible parties should consider in assessing critical pollution areas and thus working with landowners or stakeholders to devise proper control measures to eliminate the pollution problems. The determination of proper control measures (i.e., BMPs) for each of these diverse watersheds using optimization techniques and genetic algorithms are the subject of our other research project that will soon be reported to the literature.

  1. how to easily understand the curve figures such as 4, 5, or 7.

Figures 4, 5, 7, and 8 are closely related and provide information on the concentration or spread of selected nonpoint source (NPS) constituents in each watershed. To fully understand these figures, it is necessary to have a basic understanding of the Soil and Water Assessment Tool (SWAT). SWAT divides a watershed into multiple sub-watersheds, which are then further divided into hydrologic response units (HRUs). HRUs consist of the same land use, soil type, slope class, and management data, and represent the smallest calculation unit in a watershed.

Using SWAT, the output of runoff, sediment, and nutrients (N and P) can be obtained for each HRU. To analyze the figures, the NPS output is first obtained from each HRU, and the density of these outputs is then calculated by dividing the output of model to the area of each HRU, thus obtaining the density of each pollutant. Then, the densities are sorted in reverse order to obtain critical NPS hotspots. For example, in Figure 5(c), the top 10% of sorted HRUs may account for 20% of the total area in Watershed 263 (green line), but they output 25% of the total nitrogen. In contrast, 20% of the hotspot areas in Greensboro and Watts Branch watersheds account for more than 40% of the total nitrogen output. These findings suggest that the concentration of NPS constituents is more concentrated in Greensboro and Watts Branch than Watershed 263. This sorting strategy may help in directing NPS control measures (i.e., BMPs) more efficiently and effectively.

The curvature of the lines in the figures is directly related to the spatial concentration of NPS hotspots. More curved lines indicate a greater concentration of NPS hotspots. By implementing control measures (BMPs) in these sorted and concentrated hotspot areas, more NPS pollutants can be reduced in Greensboro and Watts Branch watersheds than Watershed 263. However, for watershed 263, NPS hotspots are spatially more spread-out, and that makes implementation of control measures more challenging and costly.

  1. The hotpot results lack numerical results supportance and where are the detailed comparison between the baseline and designated watershed?

In this study, we identified the most concentrated areas of NPS pollution, referred to as hotspots, by selecting HRUs that accounted for the top 20% and top 50% of NPS output. The resulting hotspots are depicted in Figures 7 and 10, with the red areas representing the top 20% of NPS output and the red and yellow areas representing the top 50%. We also presented Tables 4 and 5, which provide information on the percentage of area and NPS output density thresholds for these hotspots in each of the six watersheds studied. These thresholds were established based on NPS output density, with values above the threshold indicating hotspot areas and values below the threshold not being considered hotspots. Implementing BMPs in these hotspot areas has the potential to reduce more NPS pollution compared to other areas with lower output density.

Reviewer 3 Report

This is a well-written manuscript addressing spatial distribution of non-point source pollution due to changing land cover using the Soil & Water Assessment Tool model. The manuscript is clear, well-organized, and seems worth of publication. The methods and materials section includes pertinent details about the watershed, and landcover data, and the software. The results presented would be of interest to the audience of this journal.

That said, I have some concerns about the current manuscript which need to be addressed before the article is published.  

Major

·       The abstract needs to include significant or original contributions from this study.

·       Section 1 Introduction:

o   This section needs to include more modelling approaches including recent models to further differentiate the contribution of this study.

·       Section 2

·       Section 2.2 SWAT model parameter development in three watersheds

o  Line 204: The three parameters detailed in this section are used as the objective function. However, the objective function needs to be explicitly expressed in the methodology. I would also suggest that the author’s include further details about the modeling approach as supplementary information OR include the data and model repository.  

·       Section 3 Results & Discussion

o   I think that the paper would benefit from including assumptions of the modeling approach and a discussion about the limitations of this study.

·       Section 4 Conclusion

o   I suggest including the future work that can be carried out to improve upon the current approach.

Minor

·       Figure 2. Caption: There is a typo.

·       Figure 2 & 6. These figures need to annotate and denote the different watersheds (Greensboro, Watts or Watershed 263).  

·       Figure 6. Caption: Include the subfigure description in the caption.

Author Response

  1. The abstract needs to include significant or original contributions from this study.

This research confirms the initial hypothesis that as land cover changes from agricultural to ultra-urban across watersheds, the spatial distribution of nonpoint source (NPS) pollutant outputs becomes increasingly uniform across the landscape. The study finds that the spatial distribution of NPS pollution source areas is a function of the major land cover category of study watersheds. Therefore, control strategies should be adapted accordingly. If NPS pollution is distributed unevenly across a watershed, hotspot areas output a disproportionate amount of pollution and require more targeted and intensive control measures. On the other hand, if the distribution of NPS pollution is more uniform across a watershed, the control strategies need to be more widespread and encompass a larger area.

  1. This section needs to include more modelling approaches including recent models to further differentiate the contribution of this study.

I have added information about modeling and discussed the model selection considerations in introduction (line65-75).

  1. Line 204: The three parameters detailed in this section are used as the objective function. However, the objective function needs to be explicitly expressed in the methodology. I would also suggest that the author’s include further details about the modeling approach as supplementary information OR include the data and model repository.

Yes, we have added more detailed description about the SWAT model, SWAT model input and output, SWAT model calibration in section 2.2.

  1. I think that the paper would benefit from including assumptions of the modeling approach and a discussion about the limitations of this study.

SWAT is a widely used watershed-scale model that simulates the impacts of land management practices and climate change on water, sediment, and nutrient fluxes in a river basin. We refer the reviewers the model user manual and the original paper by developers of the model(Arnold et al., 2013; Arnold et al., 2012; Arnold et al., 1998).

  1. I suggest including the future work that can be carried out to improve upon the current approach.

Yes, I have added the assumption, limitation, and future work in the conclusion as follow.

“As for the limitations of the study. Firstly, the research findings are based on the assumption that the simulation framework SWAT can accurately represent the hydrological processes of the study watersheds. Although the models have been calibrated and validated, it is important to confirm the findings of this research with other modeling methods and field survey studies. Secondly, the incomplete daily gauging data for all three watersheds could introduce biases and affect the model performance. Therefore, it is essential to obtain complete and accurate data for future studies. Lastly, the research only focuses on three watersheds in the Chesapeake Bay Basin in the U.S. Hence, it is necessary to evaluate the findings in other areas of the U.S. or the world to determine the generalizability of the study results.

In the future, it is recommended to conduct research in other areas to test the generalizability of the current findings. Additionally, it is important to incorporate land cover information into existing Best Management Practice (BMP) allocation tools to account for the spatial distribution of NPS constituents across different land cover types. In urban areas where NPS hotspots are more spatially distributed, investigating BMP adoption behavior, and identifying effective social interventions for NPS pollution control is crucial. Furthermore, quantifying the social impacts of BMP adoption can inform effective social interventions. These recommendations aim to enhance our understanding of NPS pollution control and guide the development of more effective BMP allocation strategies.”

  1. Figure 2. Caption: There is a typo.
  2. Figure 2 & 6. These figures need to annotate and denote the different watersheds (Greensboro, Watts or Watershed 263).
  3. Figure 6. Caption: Include the subfigure description in the caption.

All suggestions have been updated in the manuscript.

References

Arnold, J. et al., 2013. SWAT 2012 input/output documentation, Texas Water Resources Institute.

Arnold, J.G. et al., 2012. SWAT: Model Use, Calibration, and Validation. Transactions of the ASABE, 55(4): 1491-1508. DOI:10.13031/2013.42256

Arnold, J.G., Srinivasan, R., Muttiah, R.S., Williams, J.R., 1998. Large area hydrologic modeling and assessment part I: model development 1. JAWRA Journal of the American Water Resources Association, 34(1): 73-89.